# The Role of MTBP as a Replication Origin Firing Factor

**DOI:** 10.3390/biology11060827

**Published:** 2022-05-27

**Authors:** Eman Zaffar, Pedro Ferreira, Luis Sanchez-Pulido, Dominik Boos

**Affiliations:** 1Molecular Genetics II, Centre for Medical Biotechnology, University of Duisburg-Essen, 45141 Essen, Germany; eman.zaffar@uni-due.de (E.Z.); pedro.ferreira@uni-due.de (P.F.); 2Medical Research Council Human Genetics Unit, IGC, University of Edinburgh, Edinburgh EH9 3JR, UK; luis.sanchez-pulido@ed.ac.uk

**Keywords:** MTBP, DNA replication, replication initiation, replication origin firing, molecular mechanisms, regulation

## Abstract

**Simple Summary:**

Copying the chromosomal DNA completely and faithfully during the process of DNA replication is key to inheriting the genetic information in an unaltered state. To achieve this, replication initiation, which generates the complex molecular machines on the chromosomes that copy the DNA, must be precisely regulated. Despite our profound understanding of bacterial and viral DNA replication, our knowledge about the molecular processes and cellular regulation of replication initiation in eukaryotic cells was surprisingly scarce for a long time. Recently, fundamental progress has been made by studying purified replication initiation proteins structurally and biochemically. The MTBP protein (Mdm2-binding protein) was the last replication initiation factor to be identified in higher multicellular eukaryotes. MTBP is the counterpart of the Sld7 protein found in the simpler eukaryote budding yeast. MTBP is essential for replication initiation in cultured human cells and is emerging as a molecular relay for signals that control replication initiation at origins, such as initiation efficiency, placement and timing. We here discuss recent progress to unravel the molecular initiation processes and the role of MTBP.

**Abstract:**

The initiation step of replication at replication origins determines when and where in the genome replication machines, replisomes, are generated. Tight control of replication initiation helps facilitate the two main tasks of genome replication, to duplicate the genome accurately and exactly once each cell division cycle. The regulation of replication initiation must ensure that initiation occurs during the S phase specifically, that no origin fires more than once per cell cycle, that enough origins fire to avoid non-replicated gaps, and that the right origins fire at the right time but only in favorable circumstances. Despite its importance for genetic homeostasis only the main molecular processes of eukaryotic replication initiation and its cellular regulation are understood. The MTBP protein (Mdm2-binding protein) is so far the last core replication initiation factor identified in metazoan cells. MTBP is the orthologue of yeast Sld7. It is essential for origin firing, the maturation of pre-replicative complexes (pre-RCs) into replisomes, and is emerging as a regulation focus targeted by kinases and by regulated degradation. We present recent insight into the structure and cellular function of the MTBP protein in light of recent structural and biochemical studies revealing critical molecular details of the eukaryotic origin firing reaction. How the roles of MTBP in replication and other cellular processes are mutually connected and are related to MTBP’s contribution to tumorigenesis remains largely unclear.

## 1. Introduction

Faithful genetic inheritance during eukaryotic cell division rests on the strict alternation between a round of complete and accurate chromosome duplication followed by exact sister chromatid segregation. Duplication of the genome is carried out by multi-protein complexes called replisomes. The process of generating replisomes at replication origins, replication initiation, needs to be both efficient and appropriately regulated to facilitate complete and accurate genome duplication. Under-initiation, over-initiation and ectopic initiation lead to incomplete replication or re-replication, genomic instability and cancer [1,2,3,4,5,6]. 

Eukaryotic replication initiation is divided into two steps, origin licensing and firing [7,8]. Licensing occurs in the G1 cell cycle phase and loads two inactive replicative helicases on double-stranded DNA in a head-to-head orientation, a structure called pre-replicative complexes (pre-RCs). Origin firing occurs in the S phase and converts pre-RCs into two mature replisomes.

The correct regulation of replication initiation underlies many fundamental characteristics of genome replication. The regulation of licensing and firing by CDK cell cycle kinases ensures that replication is limited to exactly one round per cell division [9]. High CDK activity in the S phase induces origin firing but inhibits re-licensing and, therefore, re-initiation of the fired origins so that no part of the DNA is copied more than once.

An appropriate regulation of replication initiation is also required to replicate the large eukaryotic genomes to completeness. Eukaryotic cells use multiple origins in order to fully duplicate their genomes in an appropriately short time [10]. The strategy that has evolved is to license many more potential origins than fire in a normal round of replication. Out of the many pre-RCs formed cells pick the origins to fire. Cells must avoid extremely large distances between neighboring fired origins such that the two opposing replisomes cannot copy. To avoid extra-large inter-origin distances, firing efficiency must be high enough to generate the required number of replisomes. Moreover, the distribution of the replication initiation sites must sufficiently even. If non-replicated gaps remain, the attempt to segregate the incomplete sister chromatids in the subsequent mitosis leads to mis-segregation, broken chromosomes, and genetic alterations [11].

Other, less clearly established regulations must exist. The concept that efficient firing in the S phase is central for accurate and complete genome duplication implies that initiation outside the S phase, be it ever so rare, must be suppressed, probably involving active regulation. The correct regulation of replication initiation also mediates the temporal replication program in higher eukaryotes, the reproducible order of replicating of the different chromatin domains [12]. Moreover, replicating the complex eukaryotic genomes also requires the context-dependent regulation of replication initiation to coordinate replication with other chromatin processes, for example, to avoid transcription–replication collision [13].

In addition to the above-described regulations in unperturbed cell cycles, the regulation of replication initiation is an important branch of the cellular response to DNA damage [14]. Replication origin firing becomes inhibited upon DNA damage or other sources of replication stress to avoid genetic alterations. Despite this inhibition of origin firing, a subset of origins that are dormant in unperturbed conditions fire. This dormant origin firing occurs in genomic regions that have already started replicating at the time the DNA damage signal is induced, leading to an overall higher number of origin firing events in these actively replicating regions upon DNA damage. This results in an increased number of replisomes in these regions that is thought to rescue replication of the DNA between two opposing forks stalled at DNA lesions, which helps to prevent genetic alterations. Dormant origins are recruited from the pool of excess pre-RCs, formed in the previous G1 phase.

Many of the molecular mechanisms underlying the described regulations of replication initiation remain unresolved. Those that have been investigated in more detail showed that the regulation of origin firing occurs at the step of pre-initiation complex formation (pre-IC formation), the first sub-step of activating the Mcm2−7 helicase that is inactive in pre-RCs. This involves the stable recruitment of the Cdc45 and GINS factors to the helicase to form the Cdc45-Mcm2−7-GINS (CMG) complex (details below) [15,16,17]. Mdm2 binding protein (MTBP), which was originally described as an interactor of the Mdm2 protein, is required for generating replisomes in human cells and is emerging as a regulation focus to control replication origin firing in metazoa, alongside the more classic regulation targets Mcm2−7, TopBP1 and Treslin/TICRR [18,19]. We will first describe recent progress in our understanding of the molecular processes of CMG formation and then discuss the current view of the contribution of MTBP.

## 2. Molecular Processes of CMG Formation

Recent research—mainly investigating budding yeast replication—has significantly advanced our understanding of the molecular processes of replication initiation. Especially regarding the CMG assembly step: (1) biochemical reconstitution of origin licensing and firing allowed the biochemical dissection of these processes and the isolation of intermediate assemblies of the replication initiation reaction [20,21,22,23]; (2) structural work characterizing pre-RCs and CMGs defined the start and end points of the CMG formation reaction [20,21,24,25,26].

### 2.1. Structural Studies on Pre-RCs and CMGs Outlined the Main Processes of the CMG Formation Reaction

Insight into the structure of pre-RCs showed the structure of the start point of the CMG formation process. Initially, biochemical reconstitution of origin DNA-dependent licensing with the yeast origin recognition complex (ORC), Cdc6, Mcm2−7-Cdt1, and ATP enabled the determination of the pre-RC structure at relatively low resolution by electron microscopy [20,21]. Later, pre-RCs purified from yeast G1 chromatin were used for electron microscopy at a near-atomic resolution [24]. These pre-RC structures contained two replicative MCM2−7 helicase hexamer rings (Mcm2−7 double hexamers) bound to origin DNA. Mcm2−7 belongs to the helicase family of the AAA+ ATPase type (ATPase associated with various cellular activities) [27]. The Mcm2−7 rings are in a head-to-head orientation, meaning their N-terminal domains face each other (Figure 1). The path of the double-stranded DNA (dsDNA) was not resolved in these early pre-RC structures. It was proposed that the dsDNA passes through the central channel of the Mcm2−7 double hexamer. This was the model most consistent with the observation that loaded pre-RCs could slide on dsDNA in an ATP-independent manner (as opposed to the energy-consuming DNA unwinding/translocation reaction) [20,21]. This proposed path of the DNA was later confirmed by cryo-electron microscopy [28]. In the Mcm2−7 ring, a weak interface that can open was suggested to exist between Mcm2 and Mcm5 [20,25,29].

The structure of the CMG helicase revealed major re-configurations during the formation of CMG from pre-RC. Rather than forming a double hexamer, the Mcm2−7 ring in the CMGs is a single hexamer bound to Cdc45 and GINS [25,30,31] (Figure 1). Cdc45 and GINS bridge the weak Mcm2-5 interface [25,32,33]. A stable ring is consistent with CMG’s role as the central core of the mature replisome, which must be tightly associated with the DNA template. In addition, Cdc45’s and GINS’ association with Mcm2−7 appears to activate the Mcm2−7 helicase because CMG showed robust dsDNA unwinding activity in vitro when incubated with model DNA fork substrates, whilst pre-RCs were helicase-inactive [34]. These in vitro DNA unwinding reactions were independent of origin-dependent replication initiation. The recombinant CMG helicase apparently slipped onto the ssDNA overhang to engage the dsDNA–ssDNA fork and to start unwinding. During the origin firing reaction, CMG associates with DNA polymerase epsilon to form the CMGE complex (CMG-polymerase epsilon) [26,35,36,37,38,39]. DNA polymerase epsilon has a polymerase-independent essential function during replisome formation [40,41,42,43]. In CMGE, polymerase epsilon seems to lead to poorly investigated re-configurations of the helicase and its engagement with the DNA [37]. Unlike pre-RCs, the CMGE ring encircles single-stranded DNA rather than dsDNA in vitro. This was suggested by experiments in *Xenopus* egg extracts and by in vitro reconstitution, where road blocks on the leading strand hampered the progression of replisomes and CMG. In contrast, lagging strand blocks had little effect [44,45,46]. This suggested that CMG is a leading-strand-tracking helicase that can readily bypass lagging strand blocks.

Initial evidence had suggested that the CMG helicase travels with the C-terminus facing the DNA fork [38]. Such an orientation is intuitive based on the head-to-head Mcm2−7 double hexamer structure with the C-terminal domains facing away from the interaction interface. The hexamers could simply split, move apart and form bidirectional replisomes. However, structural models built from CMG incubated with model DNA forks clearly positioned the CMG helicase with the N-terminal Mcm2−7 domains facing the DNA fork [36]. This N-terminus-first orientation was found with CMG formed by origin-dependent in vitro replication initiation [23]. The N terminus-first CMG orientation dictates that the two Mcm2−7 helicases in pre-RCs must pass each other during CMG/replisome formation to engage the fork (Figure 1). Such a passing is only possible upon melting the dsDNA trapped inside the central channels of both Mcm2−7 hexamers followed by the extrusion of opposite ssDNA strands from the channels because one remaining dsDNA-bound Mcm2−7 would block the bypass of the other CMG. This need for extrusion from both Mcm2−7 rings suggests that the passing mechanism helps ensure that replication can only start from both hexamers simultaneously, guaranteeing the bi-directionality of replication.

Together, these advances outline the re-configurations necessary for forming CMGE from pre-RCs (Figure 1): (1) The Mcm2−7 double hexamers must split into single hexamers, (2) the origin dsDNA must melt, (3) both Mcm2−7 rings must open, presumably at the Mcm2-5 gate, to extrude opposite DNA single strands from their centers, (4) the Cdc45, GINS and DNA polymerase epsilon must engage with Mcm2−7 to activate the helicase activity, (5) The CMG helicases must bypass each other to properly engage with the DNA fork.

### 2.2. In Vitro Re-Constituted Eukaryotic Replication Initiation Allowed Initial In-Detail Studies

Recently, origin-dependent replisome formation was re-constituted solely from purified components [22]. Experiments omitting individual factors showed that CDK, DDK, Sld2, Sld3-Sld7, Dpb11, Cdc45, GINS and DNA polymerase epsilon are both required and sufficient to form salt-stable CMGs from pre-RCs on origin DNA in the presence of ATP. The Subsequent formation of mature replisomes capable of efficient leading and lagging strand DNA synthesis requires Mcm10, DNA polymerase alpha/primase and other factors in vitro and in yeast cells [22,47,48,49,50].

DDK phosphorylates the Mcm2−7 helicase at multiple sites preferentially in the pre-RC configuration [51,52,53]. Pre-RC recruitment experiments in cell lysates and in vitro showed that this DDK phosphorylation is required to attract Sld3-Sld7 and Cdc45 [54,55]. This initial DDK-dependent Sld3-Sld7-Cdc45 recruitment step appears to reflect a primary unstable pre-RC binding event that precedes stable Cdc45 incorporation during CMG formation in vivo and in vitro [16,56]. Single-molecule microscopy using the in vitro re-constituted origin firing system suggested that the initial unstable association is indeed an intermediate of stable CMG formation [56]; increasing pre-RC phosphorylation by DDK at multiple sites led to the recruitment of increasing numbers of unstably bound Sld3-Sld7-Cdc45 molecules. Importantly, this increased unstable recruitment went along with elevated levels of stable CMGs. A gradual response to DDK activity levels could pose a mechanism to determine the strength of an origin and the likelihood of firing.

Whilst the initial Sld3-Sld7-Cdc45 recruitment to pre-RCs was independent of additional factors, the stable integration of Cdc45 to form CMGs required the presence of GINS, Sld2, Dpb11 and DNA polymerase epsilon [56]. CMG formation also depends on CDK, which is required to recruit all other CMG formation factors downstream of Mcm2−7-Sld3-Sld7-Cdc45 [22,54]. The only two essential CDK substrates for replication in yeast are Sld3 and Sld2 [57,58]. CDK phosphorylates Sld3 at two CDK consensus sites, which turns the region around these CDK sites into a specific high-affinity binding surface of a pair of tandem BRCT repeats (breast cancer 1 C-terminal repeat) in the N-terminus of the Dpb11 protein [57,58,59]. CDK might also phosphorylate Sld2 on several consensus sites. Phospho-Sld2 binds to the C-terminal tandem BRCT pair in Dpb11, which is strictly dependent on Sld2–threonine 84 phosphorylation [57,60,61]. In yeast cells and in vitro, Sld2-Dpb11 may form the so-called pre-loading complex (pre-LC) with GINS and DNA polymerase epsilon, recruiting these essential firing factors to pre-RCs, which are associated with Sld3-Sld7-Cdc45 in a DDK- and CDK-dependent manner [43,56]. The resulting intermediate may be considered the molecular representation of the previously proposed pre-initiation complex (pre-IC), defined to form strictly CDK-dependently [15]. Pre-ICs could be detected in yeast cells using systematic chromatin immuno-precipitation [17]. This and other studies also suggested that the in vivo situation may be more complex than in vitro, with a high degree of inter-dependence of the association of the pre-IC factors [62].

Sld3-Sld7, Sld2 and Dpb11 have not been found to be part of the mature replisome [16,63]. Consistently, they are thought to dissociate from pre-ICs during CMG formation. Therefore, their role during CMGE formation can be described as facilitating the productive integration of Cdc45, GINS and DNA polymerase epsilon into the CMGE complex.

A recent study using the in vitro origin firing system provided a detailed glimpse into the molecular processes of CMG formation [23]. One set of experiments analyzed the loading and hydrolysis of ATP using thin layer chromatography that distinguishes between ADP and ATP. In vitro CMG formation experiments involving a combination of radioactive α-^32^P-ATP and slowly hydrolysable ATP-γS suggested that CMG formation, but not pre-IC formation, requires the exchange from ADP to ATP in the active sites of Mcm2−7 in pre-RCs. The hydrolysis of ATP was not required for CMG formation but for the subsequent Mcm10-dependent step. Another set of experiments analyzed DNA unwinding in detail, using the determination of the linking number of closed DNA circles during the origin firing reaction. The DNA double-strand was found to unwind to a very limited extent during CMG formation, which was concomitant with ATP binding to the CMG, and around the time when the double hexamers split. More extensive unwinding occurs in the subsequent Mcm10-dependent step, probably alongside ssDNA strand extrusion.

### 2.3. Pre-IC Formation Is a Main Regulation Step of Origin Firing

Three characteristics make the pre-IC formation step particularly suitable for regulating origin firing: (1) pre-IC formation is the first sub-step of forming replisomes from pre-RCs. Regulating the first step of a pathway saves resources. (2) The pre-IC factors Sld2, Sld3-Sld7, Dpb11 and DDK are rate-limiting for origin firing in yeast cells, which is a prerequisite for ensuring that controlling their activities takes effect [64,65,66]. (3) It was suggested based on the highly inter-dependent binding properties of pre-IC factors to origins in vivo that pre-IC formation has switch-like properties [17,62].

Sld2, Sld3-Sld7, Dpb11 and pre-RCs have emerged as regulation foci that integrate signals from various kinase and other pathways to adapt origin firing to the cell cycle, DNA damage and chromatin contexts. The S phase-specificity of origin firing is mediated by the CDK and DDK kinases, which have low activities in the G1 phase that rise sharply at S phase entry. It has been shown that these two kinases cooperate in the cell cycle control of origin firing. Genetic manipulation in yeast cells has shown that simultaneously bypassing both CDK and DDK dependency of replication initiation induced replication in the G1 phase whereas bypassing the individual kinase pathways did not (DDK bypass) or had little effect (CDK bypass) [57]. As detailed above, the critical substrates are pre-RCs for DDK and Sld2 and Sld3 for CDK. Additionally, the DNA damage-dependent origin firing inhibition to prevent the replication of damaged templates is regulated at the pre-IC step. Central to this is the inactivating phosphorylation of Sld3 and Dbf4 by the Rad53 DNA damage checkpoint kinase [67,68,69]. Seemingly paradoxically, the Mec1 checkpoint kinase that activates Rad53 not only inhibits firing upon DNA damage but also promotes firing in normal growth conditions by phosphorylating the Mcm4 and 6 subunits of pre-RCs [70]. These phosphorylations serve as priming phosphorylations for DDK.

Recently, a picture has emerged that suggests that specific regulation mechanisms exist in yeast cells to meet the specific requirements of certain genomic regions for firing regulation. An example is replication time. At heterochromatic origins and some other late-firing origins, Rif1-PP2A removes DDK phosphorylations to mediate late firing time when compared to more efficient earlier origins [71,72,73]. At pericentromeric heterochromatin, the kinetochore protein Ctf19 is required to recruit DDK to mediate the exceptionally early firing of pericentromeric origins [74]. Recently, Sld3 was found to interact with the histone acetyl transferase Esa1 at the silent HML alpha locus. This interaction controls both origin firing time and transcriptional repression of the locus [75]. Intriguingly, origin firing factors must also be controlled outside the S phase. In mitosis and the G1 phase, Sld2 levels are attenuated by degradation and other mechanisms to prevent genetic instability and to maintain origin firing time [60,64,65,66,76,77]. Moreover, Sld3 phosphorylation by checkpoint kinases in G2 was found to help prevent re-replication [78]. For the sake of brevity, we refer for details to the papers cited in our recent review on origin firing regulation [79].

## 3. The MTBP Origin Firing Factor

### 3.1. The Yeast MTBP Orthologue Sld7 Has a Little-Defined Important Role in Origin Firing

Sld7 was identified in a genetic screen in budding yeast for synthetically lethal genes with Dpb11 (Sld) [80]. Sld7 binds to the N-terminal region of Sld3 throughout the cell cycle [80,81]. Both the Sld3 N-terminal region and Sld7 were not essential for yeast survival, but Sld7 was required for the normal progression of replication and for resistance against low doses of hydroxyurea, consistent with an important function during genome replication [80]. We know little about the molecular activities of Sld7 that are relevant for replication. A Sld7–-Sld3 dimer can be purified to homogeneity. The heterodimer is co-recruited to pre-RCs DDK-dependently in vitro and within cells [22,55,66]. In vitro, origin-dependent replication is undiminished in the absence of Sld7 [55]. This suggests that yeast Sld7 is not essential for the biochemical replisome generation process per se, but has a more indirect role in replisome formation in cells. However, it needs to be noted that the situation in yeast might be special because Sld7 is important, but not essential, for cellular replication, whereas MTBP appears to be absolutely required [19,80]. One molecular role of Sld7 could be to stabilize Sld3, as suggested by Sld3 protein levels decreasing in *sld7**Δ* cells [80]. A crystallographic study suggested that the C-terminal Sld7 region is a homodimerization domain [81]. Although the in vivo relevance of Sld7 homodimerization remains untested, it is tantalizing to speculate that Sld7 dimerization couples the activation of the two Mcm2−7 helicases in pre-RCs in order to avoid the situation that only one replisome is built.

### 3.2. The Pre-IC Factor Complex MTBP–Treslin/TICRR–TopBP1 Is Required for S Phase-Specific Origin Firing in Metazoa

Many fundamental aspects of the molecular processes and cell cycle regulations discussed above for yeast are conserved in metazoa, including aspects involving MTBP-Treslin/TICRR, metazoan Sld7-Sld3. The set of core factors sufficient for pre-IC formation defined by biochemical re-constitution in yeast are conserved in the higher eukaryotes [22]. In metazoa, these factors are Mcm2−7, DDK, CDK, MTBP, Treslin/TICRR, TopBP1 (Dpb11-homologous), RecQL4 (Sld2), GINS, Cdc45 and DNA polymerase epsilon. Treslin/TICRR, MTBP and TopBP1 are required for replication origin firing [19,82,83,84,85,86]. Similar to their yeast counterparts, Treslin/TICRR, MTBP and TopBP1 form an essential protein complex in metazoan cells under the control of the cell cycle CDK kinase to mediate the S-phase specificity of origin firing [19,87,88]. Treslin/TICRR is phosphorylated on two CDK consensus sites, and this phosphorylation promotes the interaction with the N-terminal triple BRCT repeat domain of TopBP1, which constitutes a phosphorylation-dependent peptide interaction unit [59,87,88]. The Treslin/TICRR CDK sites and the TopBP1 BRCTs show sequence homology with their counterparts in Sld3 and Dpb11, respectively [82,88]. Inactivating mutations of both the CDK-dependent interaction surface of Treslin/TICRR and the TopBP1 triple BRCT domain suppressed the replication in cells or *Xenopus* egg extracts, respectively [86,87,88]. Conversely, phospho-mimetic mutations of the Treslin/TICRR CDK sites reduced the duration of the S-phase and increased DNA synthesis, consistent with increased origin firing [89]. The control of Treslin/TICRR, MTBP and TopBP1 by the DDK kinase may also be conserved in metazoa. DDK was suggested to strengthen the interaction of Treslin/TICRR-MTBP with pre-RCs, reminiscent of the DDK-dependent recruitment of Sld3-Sld7 to pre-RCs in yeast [90]. DDK also promoted the interaction of Treslin/TICRR-MTBP with TopBP1 [90]. A similar mechanism was not reported in yeast. In yeast, Sld2 is also an essential CDK target for replication initiation [60,61]. The metazoan orthologue of Sld2 is RecQL4, based on the sequence homology of their N-terminal domains [91,92,93]. RecQL4 is required for replication in *Xenopus* egg extracts and vertebrate cells [91,92,94]. However, two arguments speak against RecQL4 having retained a function in mediating the CDK-dependence of firing in all metazoan organisms: (1) RecQL4 phosphorylation was found to be not required for TopBP1 interaction [91], and (2) the tandem BRCT repeat domain in TopBP1 that is conserved with the Sld2-binding BRCT in Dpb11 is not essential for DNA replication in *Xenopus* egg extracts [86]. In contrast to vertebrates, *C. elegans* SLD-2 has retained its CDK-dependent binding to the TopBP1 orthologue Mus-101, as well as its cellular function of coupling replication to the S phase [95].

These findings indicate that, despite fundamental conservation, metazoa have evolved specific aspects of the origin firing process, presumably to accommodate the more complex requirements of their vast genomes. One such metazoan-specific mechanism might be the involvement of biomolecular condensates in the control replication initiation. It was recently shown that a specific type of intrinsically disordered regions is present in several metazoan origin licensing factors that have the capability to promote biomolecular condensate formation [96]. The condensates concentrated licensing factors, promoted pre-RC formation and were inhibited by CDK phosphorylation, presumably as a means to dampen licensing outside late mitosis/G1 phase. Additionally, the metazoan origin firing factors possess intrinsically disordered regions, for example Treslin/TICRR and MTBP. It is tempting to speculate that biomolecular condensates help organize and control, in both space and time, replication initiation at the level of cellular replication domains. The fact that metazoans have evolved partly specific replication initiation mechanisms raises the possibility of as yet unidentified players of metazoan origin firing.

### 3.3. Roles and Characteristics of the MTBP Origin Firing Protein

MTBP was recognized as an origin firing factor due to its interaction with Treslin/TICRR [19]. MTBP’s remote but statistically significant sequence homology with Sld7 was described only later [97,98]. The Sld7-homologous region comprises parts of the N-terminal half of the protein and the extreme C-terminal region, termed the Sld7–MTBP N-terminal domain (S7M-N) and the Sld7–MTBP C-terminal domain (S7M-C), respectively. The S7M-N and C domains flank a metazoa-specific central region (Figure 2A).

RNAi in cultured human cells proved that MTBP is essential for genome replication, specifically for replisome formation, but neither for pre-RC formation nor replication elongation [19]. All three main MTBP regions, S7M-N and C domains, as well as the metazoa-specific central region, are required for MTBP’s replication-promoting activity because deletion and point mutants in these domains were incapable (S7M-N) or compromising (S7M-C, center) in supporting replication in human cells and *Xenopus* egg extracts [97,98].

The MTBP S7M-N domain is required to bind Treslin/TICRR [97]. Both proteins dimerize throughout the cell cycle [19]. Surprisingly, the S7M-N domain turned out to be part of a beta-barrel fold that is structurally similar to the beta-barrel folds in Ku70 and Ku80 that mediate Ku70-Ku80 dimerization (Figure 2) [99,100]. A similar Ku70/80-homologous beta-barrel fold is present in the M domain of Treslin/TICRR [99]. Mutating the Treslin/TICRR beta-barrel showed that it is required to bind MTBP [99]. Together, these findings suggested two conclusions, (1) that MTBP/Sld7 and Treslin/TICRR/Sld3 are related by structure and (2) that they form a Ku70-Ku80-like heterodimer (Figure 2B). Replacing Treslin/TICRR or MTBP in cells with non-dimerizing mutants by RNAi did not allow genome replication, underscoring that MTBP and Treslin/TICRR act as a complex during replication origin firing [19,97]. This was supported by work in *Xenopus* egg extracts, where the immuno-depletion of MTBP or Treslin/TICRR co-depleted the other protein and suppressed replication [90,98]. Replication in the extract could not be reconstituted by adding back the purified individual proteins but by adding the Treslin/TICRR–MTBP complex.

The S7M-C domain of MTBP may constitute an MTBP dimerization domain. This was first suggested by the observation that the S7M-C domain of Sld7 can self-interact [81]. Sld7 homodimerization may help ensure that the Mcm2-7 helicases in pre-RCs are activated. Treslin/TICRR and MTBP were recently reported to form dimers of heterodimers in *Xenopus* egg extracts [90]. The MTBP–S7M-C domain is important for replication, as shown by RNAi-replacement experiments in Hela cells. MTBP mutants with a deletion of the S7M-C domain (MTBP–ΔS7M-C) supported replication to only about 50% of MTBP–WT [97]. Experiments that forced MTBP–ΔS7M-C to dimerize were consistent with the S7M-C domain promoting replication by MTBP dimerization: fusing a self-interacting GST tag to MTBP–ΔS7M-C rescued the capability of MTBP–ΔS7M-C to support replication, whereas a non-dimerizing GFP tag had no effect [97]. However, a biochemical activity of S7M-C to support the formation of dimers of MTBP-Treslin/TICRR heterodimers was not reported. Another role of S7M-C was suggested. The domain showed an in vitro DNA binding activity with a preference for G4 quadruplex structures [98]. This led to the proposal that the MTBP–S7M-C domain may bind G4-containing OGRE sequences found in origins [101,102].

The molecular role of the metazoa-specific central MTBP domain in origin firing is unclear. It contains a well-conserved region in MTBPs (but not Sld7) with binding activity for Cdk8/19-cyclin C [97]. This binding probably plays a role in replication but is not essential for origin firing (details below). Larger deletions in the central region that eliminated the Cdk8/19-cyclin C domain and adjacent regions led to a severe reduction in the ability of MTBP to promote replication in cells, suggesting that the MTBP central domain may contain unidentified replication-relevant activities [97].

Together, although the function of MTBP as an important origin firing factor has been firmly established and its domains partly characterized its molecular role during pre-IC formation awaits clarification.

### 3.4. Phosphorylation of MTBP by DNA-Damage-Signaling Kinases

Gel shift and phospho-proteome studies suggested that MTBP is phosphorylated by the ATM/R and Chk1/2 DNA damage checkpoint kinases in human cells and *Xenopus* egg extracts [103,104,105,106,107].

MTBP gel mobility shifts in *Xenopus* egg extract samples occurred without exogenously induced DNA damage [104]. A combination of pharmacological inhibition of ATM/R and Chk1, but not individual treatments, prevented the MTBP gel shift. These observations suggested the cooperation of the ATM/R and Chk1 kinases to phosphorylate MTBP in the absence of exogenous DNA damage. This may reflect truly damage-independent kinase signaling or signaling as a result of adding sperm DNA carrying low DNA damage levels to the egg extract.

The in vivo effects of MTBP checkpoint phosphorylation were investigated using MTBP phospho-site mutants in RNAi-replacement experiments in Hela cells. Phospho-mimetic aspartate mutations of four ATM/R and ten Chk1/2 sites (MTBP-14D) strongly inhibited MTBP’s capability to promote origin firing, in contrast to the corresponding non-phosphorylatable alanine mutations [103]. Only the combination of the phospho-mimetic ATM/R and Chk1/2 mutations had the full inhibitory effect. Checkpoint kinase site phosphorylation may inhibit the dimerization activity of MTBP because forced dimerization of the MTBP-14D mutant using GST-fusion revived the mutant’s replication-inducing ability. MTBP phosphorylation by DNA damage signaling may help inhibit origin firing in DNA damage conditions. However, expressing the MTBP-14A non-phospho mutant did not bypass origin firing inhibition by ionizing radiation [103]. This suggested that other signaling pathways may be sufficient for this inhibition [108]. In yeast, origin firing inhibition upon DNA damage involves the inhibition of the Sld3 and Dbf4 origin-firing-promoting pathways [67,69]. The evolution of parallel pathways may reflect the need for stringent control.

It has long been a prominent concept in the field that checkpoint signaling controls origin firing in normal cell growth conditions [109]. Indeed, the checkpoint kinase phosphorylation of MTBP may attenuate origin firing in unperturbed conditions. MTBP-14A expression decreased inter-origin distances in exponentially growing Hela cells [103]. This could reflect a role of checkpoint signaling in establishing the right origin firing frequency that is required to prevent genetic instability. However, a contribution of the basal replication stress found in cultured cancer cells could not be excluded.

### 3.5. Phosphorylation of MTBP by Cell Cycle and Other Kinases

Circumstantial evidence suggested that MTBP phosphorylation at CDK consensus sites may promote replication origin firing; MTBP has six such sites. A phospho-mimetic mutant of MTBP with all CDK sites in MTBP changed to aspartate (MTBP–CDK6D) reduced the average inter-origin distance in RNAi-replacement experiments in Hela cells, suggesting a higher firing frequency [103]. A corresponding non-phospho alanine mutant (MTBP–CDK6A) had no such effect. In which contexts this regulation applies remains unclear. CDK site phosphorylation does not seem to control origin firing genome-wide because MTBP–CDK6A did not decrease the inter-origin distances in DNA combing experiments, which would be expected if a majority of origins were controlled by CDK site phosphorylation. Therefore, CDK site phosphorylation may affect a subfraction of origins.

MTBP–CDK site phosphorylation may be executed by Cdk1 and/or Cdk8/19-cyclin C kinase (Cdk8 and 19 are very similar paralogues) [103]. Cdk8/19-cyclin C was found to stably interact with MTBP dependently on the metazoa-specific central MTBP domain [97]. When a Cdk8/19-cyclin C non-binding MTBP mutant replaced endogenous MTBP in Hela cells, a slightly slower replication rate was observed in experiments measuring bulk DNA synthesis using BrdU. The cells also showed signs of replication stress, such as increased fragile chromosome site expression. These observations suggested compromised replication in cultured human cells in the absence of an interaction between MTBP and Cdk8/19-cyclin C. Surprisingly, Cdk8/19-cyclin C phosphorylates MTBP not only at the CDK consensus sites but also at non-CDK consensus sites, suggesting that that the MTBP–CDK6A mutant may not, or not completely, eliminate the effect of Cdk8/19–cyclin C on MTBP [103]. In fact, the critical substrate of Cdk8/19-cyclin C in replication may be either MTBP itself or other replication factors. Treslin/TICRR was reported as a CDK8/19-cyclin C substrate in proteomic studies [110,111]. The cellular role of Cdk8/19-cyclin C in genome replication remains elusive. DNA combing did not detect a significant change of inter-origin distances in cells expressing a Cdk8/19-cyclin C non-binding MTBP mutant, suggesting that the interaction with the kinase does not control firing genome-wide, but may instead have local effects [103]. It is tempting to speculate that the binding of MTBP to Cdk8/19-cyclin C is related to the role of the kinase as the so-called kinase module of the mediator of transcription complex [112]. As such, the MTBP-Cdk8/19-cyclin C complex could help coordinate replication with transcription, which is necessary to preserve the genetic information [13]. This hypothetical relation with mediator-related functions of the kinase must be indirect, because Cdk8/19-cyclin C forms distinct complexes with MTBP-Treslin/TICRR-TopBP1 and core mediators [97].

MTBP was found to bind to the Plx1 kinase in *Xenopus* egg extracts [104]. Plx1 is known to have promoting effects on origin firing in DNA damage conditions and during adaptation from DNA damage signaling [113,114]. In the new study, Plx1 immuno-depletion decreased the frequency of replication initiation events in the absence of induced DNA damage, as measured by DNA fiber analysis, suggesting that Plx1 promotes replication initiation in unperturbed growth conditions [104]. It was proposed that Plx1 depletion may lower origin firing by decreasing the release of Treslin/TICRR-MTBP from replication origins. Because Treslin/Sld3-MTBP/Sld7 belong to the set of factors that are limiting for replication initiation in yeast and *Xenopus* embryos, their recycling from early firing to late firing origins is believed to be required to replicate the genome effectively and with the right timing [64,65,66,115]. Plx1 immuno-depletion increased rather than decreased the phosphorylation status of MTBP and Treslin/TICRR in the egg extracts [104]. This suggested that MTBP and Treslin/TICRR are phosphorylated upon Plx1 depletion by other kinases to inhibit firing. It is conceivable that such MTBP-phosphorylating kinases are DNA damage response kinases because these are known to inhibit origin firing, and because the ATM/R and Chk1/2 checkpoint kinases were found to phosphorylate MTBP and Treslin/TICRR [103,104,116]. Thus, Plx1 depletion may lead to MTBP phosphorylation by directly or indirectly activating DNA damage checkpoint signaling. However, the DNA damage status in Plx1-depleted *Xenopus* extract was not addressed [104].

The authors further proposed that the Plx1 activation of replication initiation involves Plx1 phosphorylation of Rif1 on serine 2058. A phospho-mimetic mutant of this site inhibited the binding of PP1 to Rif1. Rif1 suppresses origin firing by counteracting DDK [72]. Relevant DDK sites targeted by Rif1-PP1 are in pre-RCs [71,72,73]. However, also MTBP and Treslin/TICRR may be DDK substrates because phospho-dependent MTBP and Treslin/TICRR gel shifts were reversed by the pharmacological inhibition of DDK [104]. It is conceivable, albeit untested, that this DDK phosphorylation is part of the reported positive effect of DDK on the pre-RC binding of MTBP-Treslin/TICRR or on their interactions with TopBP1 [90]. If this is true, the DDK phosphorylation of MTBP is involved in the coupling mechanisms of origin firing to the S phase. In light of these potential DDK effects on MTBP and Treslin/TICRR, it is conceivable, albeit highly speculative, that Rif1 inhibits origin firing in part by reversing DDK phosphorylations on MTBP and Treslin/TICRR. Dephosphorylation would decrease complexation with TopBP1 and recruitment to pre-RCs.

## 4. Concluding Remarks

MTBP is an essential player in metazoan replication origin firing, although the molecular roles of MTBP and its yeast orthologue Sld7 remain elusive [19,98]. MTBP has earned a place among the classic firing-regulating pre-IC factors Treslin/Sld3, TopBP1/Dpb11, Mcm2–7 and RecQL4/Sld2. These proteins are limiting for origin firing, as expected for regulators [64,65,66,115]. The partial degradation of MTBP-Treslin/TICRR observed in the S phase may play a part in them becoming limiting [117,118]. Apart from this regulation by degradation, MTBP is controlled indirectly through its interaction with Treslin/TICRR and by kinase phosphorylation [97,103,104]. Although the cellular roles of these MTBP regulations remain mostly speculative, functions in coupling replication to S phase (DDK), in controlling the frequency of origin firing genome wide (damage-independent ATM/R, Chk1/2 activities, Plx1), in mediating context-specific regulations (CDK site phosphorylation, Cdk8/19-cyclin C), and in inhibiting firing in DNA damage conditions (ATM/R, Chk1/2) are plausible.

MTBP is involved in other cellular processes, such as the spindle assembly checkpoint, the actin cytoskeleton, the regulation of Myc-dependent transcription and the degradation of Mdm2 [119,120,121,122,123]. An interesting aspect is how the replication-dependent and -independent MTBP functions are linked with the roles of MTBP in tumorigenesis [121,122,124,125].

## Figures and Tables

**Figure 1 biology-11-00827-f001:**
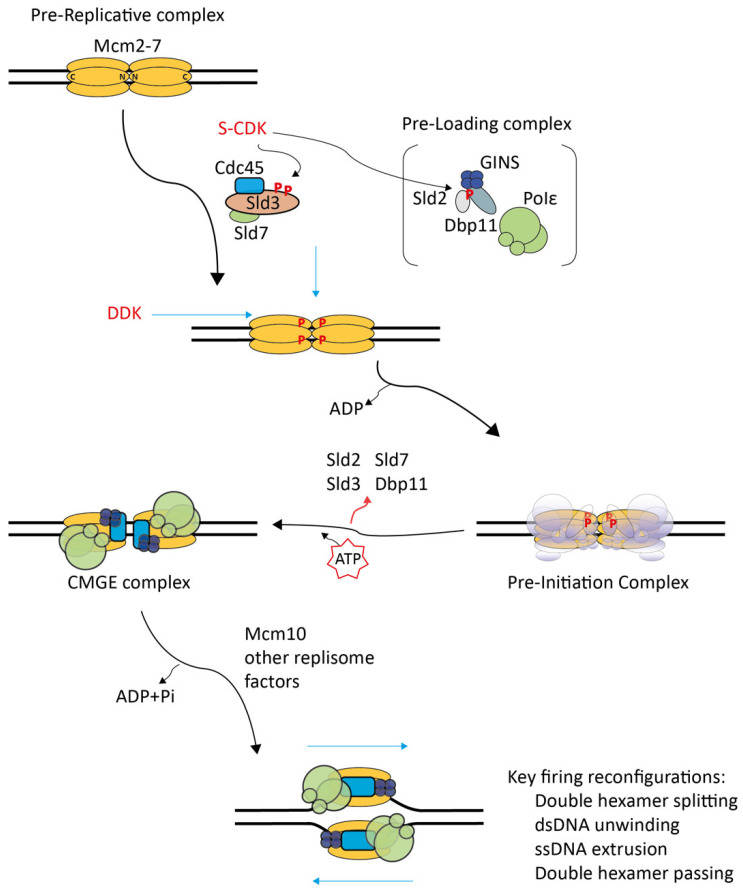
Schematic of eukaryotic replication initiation based on the budding yeast model. The first replication initiation step in the G1 cell cycle phase, licensing, loads the helicase-inactive pre-replicative complex (pre-RC) consisting of a head-to-head (N-domain to N-domain) Mcm2−7 double hexamer onto dsDNA. The second step in the S phase, origin firing, forms replisomes. DDK phosphorylates pre-RCs, recruiting Sld3-Sld7 and Cdc45. CDK phosphorylates Sld3 and Sld2, which then bind Dbp11. Sld2 forms the pre-loading complex with GINS and DNA polymerase ε (Pol ε) (brackets indicate uncertainties about complex arrangement). DDK-dependent pre-RC-recruited Sld3-Sld7, Cdc45, and pre-LC constitute the pre-initiation complex (low-lighting indicates structural uncertainties). Sld3-Sld7, Sld2 and Dbp11 dissociate, leaving behind the active CMGE helicase complex (Cdc45–Mcm2−7–GINS-DNA pol ε). Subsequent recruitment of other replisome factors completes the replisomes. The key molecular re-configurations to form replisomes from pre-RCs are the splitting of Mcm2−7 double hexamers, the melting of the dsDNA, the opening of the two Mcm2−7 rings to extrude the future lagging ssDNA strand, and the passing of the two CMGE helicases.

**Figure 2 biology-11-00827-f002:**
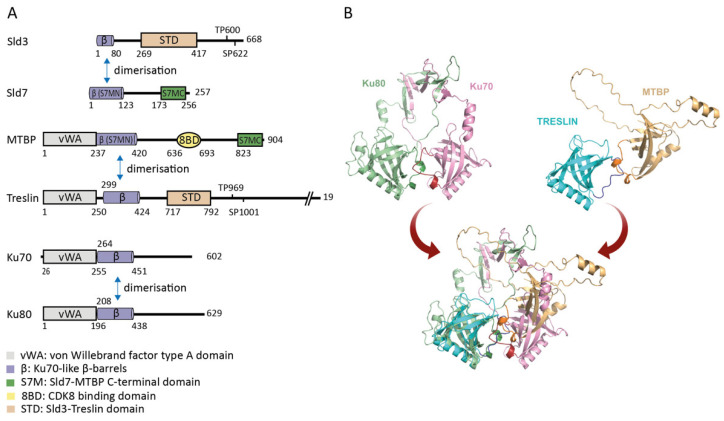
Structure of MTBP–Sld7 and related proteins (**A**) Domain architecture of Sld3/Treslin, Sld7/MTBP sharing a dimerizing Ku70-like β-barrel (β) and vWA (von Willebrand factor type A domain, Treslin/TICRR and MTBP only) domains. Protein length and amino acid position are indicated by corresponding numbers. CDK sites for Treslin/TICRR (positions 969 and 1001) and Sld3 (positions 600 and 622) are indicated. (**B**) Similar modes of interaction in Ku70-Ku80 (pbd 1JEY) and MTBP-Treslin/TICRR heterodimers are suggested by structural prediction of the MTBP-Treslin/TICRR dimer. The prediction was made using Alphafold2-advanced (Google Colab). The top models show the individual heterodimers that were superimposed to generate the bottom panel. Darker colors indicate equivalent loops between two beta-strands (b3-b4) of MTBP, Treslin, Ku70 and Ku80 that form intimate contacts. The predicted structure suggests that a different relative orientation of the β-barrels in MTBP-Treslin/TICRR may exist, but this prediction needs further experimental clarification.

## Data Availability

Not applicable.

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
