# Peer review of "The Role of MTBP as a Replication Origin Firing Factor"

_biology, 2022, doi:10.3390/biology11060827_

Round 1

Reviewer 1 Report

MTBP is a replication factor required for firing of replication origins in metazoan cells. It seems to be a counterpart of Sld7 of budding yeast, which forms a tight complex with Sld3 functioning for recruitment Cdc45 to replication origins. Cdc45, Mcm2-7 and GINS comprise the CMG replication helicase. While the Sld7 protein stabilizes the Sld3 protein, it is dispensable for cell growth. On the contrary, MTBP is essential, suggesting some more functions reside in MTBP.

The authors have been studying on MTBP and revealed several functions of this protein. In this review, they explain the initiation steps of DNA replication in eukaryotic cells and MTBP well. I do not have reservation for publication of this manuscript. Instead, I have recommendation to improve it.

The authors first explain the model based on yeast analyses and further extend it to metazoan replication focusing on MTBP. Although there is description “mainly investigating budding yeast replication” at line 100, audience may misunderstand that many results and the model are obtained from metazoan analysis. The authors have better describe it clearer. For example, the legend to Fig.1 should say it is a model based on yeast analyses

(typo) line 382, cyclion should read cyclin.

Reviewer 2 Report

Comments on Zaffar et al. ‘The role of MTBP as a replication origin firing factor’

This review focuses on the function of MTBP (Mdm2-binding protein) protein in DNA replication, especially the firing of replication origins. The manuscript is from Dr. Dominik Boos group, who are vigorously elucidating Treslin-MTBP function in vertebrates. The manuscript includes all crucial findings on these factors and their orthologues, as readers can expected. I have noticed that the manuscript contains some minor issues. Therefore, if such minor issues are fixed, I recommend the publication of this manuscript in Biology.

Minor points

  1. I saw the expression ‘initiation of origins’ many times. This does not sounds good for me. I also noticed that the word ‘initiation’ is often used alone instead of ‘replication initiation’. As authors list in keywords, ‘replication initiation’ and ‘origin firing’ are popular and appropriate expression, I think.

In the introduction, Authors describe like ‘initiation is divided into two steps, origin licensing and firing’ (Line 51). However, in the latter part of introduction, it seems that ‘initiation’ is used instead of ‘firing’. This might be confusing.  

  1. Related to 1, I also noticed that weird expressions in the introduction and the following part. For example,

              Line 53: a structure called pre -RC… Is the pre-RC a structure?

              Line 56: cell cycle CDK kinases

              Lines 66-67: … firing efficiency must be high enough to generate a sufficient number of origins in each S phase.

              Lines 82-84: difficult to get the points

              Line 105: the CMG reaction

              etc.

I think authors can make the introduction better to understand.

  1. Other points

              Line 127: ‘strengthen, the weak Mcm2-5 interface’ Is this proved? if it is, please cite the paper.

Lines 151-152: ‘CDK-dependently pre-RC-recruited Sld3-Sld7, Cdc45, and pre-LC constitute the pre-initiation complex.’ The recruitment of Sld3-Sld7-Cdc45 is not depend on CDK.

Lines 200-201: Dpb11 is missing.

Lines 205 & 208: CDK might phosphorylates all of consensus sites.

Line 238: ‘the most upstream step’  What does this mean?

Line 251: ‘than bypassing the individual kinase pathways’  The bypass of DDK cannot promote DNA replication in G1.

Lines 253-254: I do not agree. Recruitment of Sld3-Sld7-Cdc45 onto pre-RC is not identical to the pre-IC formation.

Line 316: ‘TopBP1’  Is TopBP1 under the regulation of DDK?Line 394: ‘MTBP gel shifts’  Is a mobility shift in gel?

Lines 466-467: Is it true in Xenopus egg?

Reviewer 3 Report

This paper presents a detailed review of the role of Mdm2-binding protein (MTBP) as a factor required for the firing of the origins of DNA replication in eukaryotes. This is a complex process involving the sequential interactions of many proteins. These has been a large amount of research performed so far, involving a wide range of techniques and approaches, particularly genetic studies, in vitro experiments using crude extracts, as well as in vitro reconstitution experiments with purified proteins. This paper is very well-written and covers the various steps of the process and the various types of experiments thoroughly and with understanding. The authors first describe the current research on the processes of the recruitment of the Cdc45and GINS factors to the helicase to form the Cdc45-Mcm2-7-GINS complex (CMG formation) and then discuss the role of the Mdm2-binding protein (MTBP). They cover the work on the MTBP homolog in budding yeast, Sld7. They keep track of what has become a dizzying array of participants in the process.  They compare similar proteins such as Treslin and Ku70/80. They review in detail what has been published on the activities of MTBP in DNA replication initiation and in DNA repair. Finally, however, they note that with all that has been uncovered, the molecular roles of MTBP and Sld7 remain elusive. I enjoyed this manuscript and I recommend publication.
